# The Influence of the Modernization of the City Sewage System on the External Load and Trophic State of the Kartuzy Lake Complex

Jolanta Grochowska * and Renata Tandyrak

Department of Water Protection Engineering and Environmental, Faculty of Geoengineering,
University of Warmia and Mazury in Olsztyn Microbiology, 10-719 Olsztyn, Poland; renatat@uwm.edu.pl
* Correspondence: jgroch@uwm.edu.pl

**Abstract:** A study was carried out in the Kartuzy lake complex, which has been a receiver of raw domestic sewage since the 1950s. In 2018, the city's sewage system of Kartuzy was modernized. An analysis of the water quality prior to the modernization of the sewage system revealed that the total phosphorus (TP) load that was introduced to the individual lakes from external sources substantially exceeded the dangerous load concentration (defined by Vollenweider) that causes accelerated eutrophication. The annual TP load introduced to the analyzed lakes in 2017 exceeded the critical load by 200% (Mielenko) to 1000% (Klasztorne Duże). Protective measures reduced the external loading of nutrients. In the case of Mielenko Lake, a 37% decrease in the external TP load was noted, and also a 32% decrease in the external TP load in Karczemne Lake, a 66% decrease in Klasztorne Małe Lake and a 54% decrease in Klasztorne Duże Lake was noted. The protective measures resulted in a slight decrease in the concentrations of phosphorus and nitrogen in the water. However, these changes did not improve the environmental conditions in the lakes. In a situation where the internal fertilization process in the lakes has started, the improvement of water quality will only be possible through restoration efforts with methods adjusted to the individual characteristics of each lake.

**Keywords:** nutrients; external load; lake protection; degradation

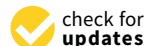

## 1. Introduction

The inflow of various chemical compounds from a catchment determines the eutrophication status of lakes, which is a natural and long-lasting phenomenon leading to their degradation [1,2]. The rate of this process and the concentrations of the main elements responsible for the increase in water trophic state can be influenced by many factors, such as water dynamics, hydrology, morphometry, and basin usage [3,4]. Human activities, such as urbanization, industrialization, or agricultural use of areas adjacent to water bodies, accelerate eutrophication and can cause the degradation of water quality [5,6]. Lakes situated in urban and agricultural areas are particularly endangered, as they hold the role of receiving water from municipal, industrial and precipitation wastes and nutrients from leaky septic tanks and agricultural fields (such as mineral fertilizers, plant protection products, pesticides). The load of nutrients introduced by sewage can be high that it disturbs the biological balance and disrupts the biogeochemical processes of an ecosystem [7–9]. Nutrients flowing into the aquatic ecosystems participate in various processes, collectively referred to as the circulation of matter. One of the main processes is the uptake of the dissolved mineral forms of nitrogen and phosphorus by primary producers for photosynthesis and their subsequent processing and release in the food web [10]. Moreover, biogenic elements in the form of organic compounds, suspensions and complexes with iron, aluminum and calcium are subject to sedimentation and accumulation in bottom sediments. The withdrawal of nutrients to bottom sediment is temporary [11,12]. A significant portion of the substances

accumulated in the sediment are processed by living organisms and returned to circulation. This process is called "internal loading" and is common in degraded lakes.

In most aquatic ecosystems, the "internal loading" in the long term has a negative value, and in degraded water bodies, as a result of the significant accumulation of nutrients in sediment and exceeding the limits of the permissible external load for a reservoir, it becomes positive. This is the stage where the transport of elements from bottom sediment exceeds their precipitation. This phase is known as saprotrophy, i.e., the moribund state of the lake whereby it is unable to transform the excess organic matter and minerals within its structure and the organisms functioning in it.

Disruption of the normal functions of a lake ecosystem is manifested primarily by massive algae and cyanobacterial blooms as well as oxygen deficits in the deeper parts of the reservoir [1]. The effect of an excessive increase in the fertilization levels of a lake is also elimination of the water body as an object utilized for communal and recreational purposes. The issue of accelerated eutrophication has generated a need for new remedy measures consisting of the protection of endangered reservoirs or restoration of those that have already been degraded [13]. Cooke et al. [14], Klapper [15], and Łopata et al. [16] provided many solutions that allow for the elimination or reduction of external sources of nutrients (protection of lakes) through technical (artificial aeration, removal of bottom sediments, selective removal of hypolimnion waters), chemical (phosphorus inactivation, sediment treatment) and biological (biomanipulation) methods carried out within a lake basin (lake restoration). Experiments conducted worldwide have proven that protective methods implemented within a catchment play the most important and essential role in improving the condition of the aquatic environment [17,18]. The research of many authors [19–21] has shown that in the case of lakes where the internal fertilization of nitrogen (N) and phosphorus (P) water from bottom sediments has started, limiting the external load does not improve the water quality. It should be emphasized once again that, for all types of restoration techniques, an important prerequisite for success and achieving long-term effects is the elimination of the underlying sources of undesirable water quality by sufficiently reducing external P loading [22,23]. These activities are arduous, time-consuming and less effective but are needed before expensive reclamation procedures can be implemented.

Mielenko, Karczemne, Klasztone Małe, and Klasztorne Duże (of the Kartuzy lake complex) are situated in the town of Kartuzy (Eastern Pomeranian District, Poland). Since 1956, these lakes have been transformed into sewage receivers for storm and municipal sewage [24].

The lake complex in Kartuzy is a river–lake system consisting of several lakes with different morphometric features and different hydraulic retention times, each showing varied degrees of degradation, which is influenced by the amount of introduced pollutants. The aim of the study is to present the response of these various water quality parameters to reduced external loading by analyzing changes in nutrient concentrations (N and P) and other trophic status indicators (chlorophyll, Secchi disc visibility).

## 2. Methodology

### 2.1. Characteristics of the Research Objects

The Kartuzy lake complex is situated in Kashubian Lakeland in the town of Kartuzy [25] (Figure 1).

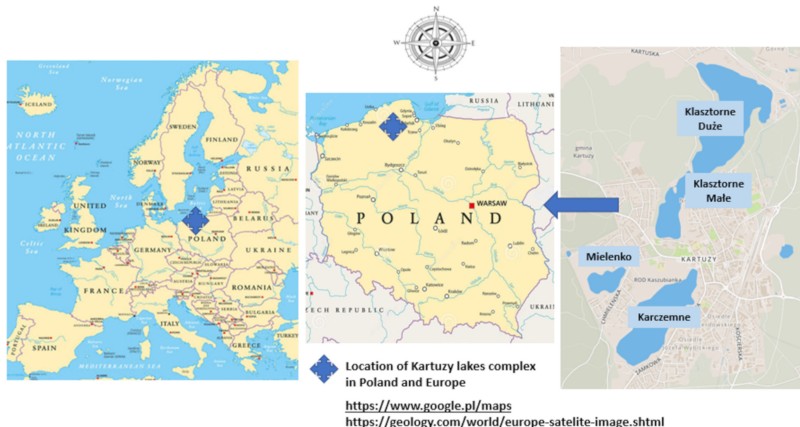

**Figure 1.** The position of Kartuzy lake complex in Poland and Europe.

Mielenko (1.9 m) and Karczemne (3.2 m) are shallow, polymictic lakes that mix to the bottom throughout the year. The basins of both lakes are semicircular (DI—0.68, DI—0.62) and poorly seated in the ground, which is depicted by their relative depths: 0.0068 and 0.0050. The following two reservoirs, Klasztorne Małe (20.0 m) and Klasztorne Duże (8.5 m) are classified as holomictic lakes that circulate twice a year. The basin of Klasztorne Małe Lake, with a shape that resembles a paraboloid DI (depth indicator) (Table 1). In contrast, the basin of Lake Klasztorne Duże, which is shallow, even, and resembles a hemisphere (DI—0.59), is moderately seated in the ground, which is reflected in the value of relative depth at 0.0112 (Table 1).

**Table 1.** Morphometric data of studied lakes [26].

| Parameter | Mielenko | Karczemne | Klasztorne Małe | Klasztorne Duże |
|---|---|---|---|---|
| Geographical coordinates | 54°19′55″ N 18°10′55″ E | 54°19′42″ N 18°11′27″ E | 54°20′21″ N 18°11′35″ E | 54°20′52″ N 18°12′10″ E |
| Height above mean sea level (m AMSL) | 204.0 | 207.3 | 203.0 | 202.3 |
| S (surphace) (ha) | 7.8 | 40.4 | 13.7 | 57.5 |
| $D_{max}$ (maximum depth) (m) | 1.9 | 3.2 | 20.0 | 8.5 |
| $D_{mean}$ (mean depth) (m) | 1.3 | 1.98 | 8.1 | 4.8 |
| $D_{relative}$ (relative depth) | 0.0068 | 0.0050 | 0.054 | 0.0112 |
| DI (depth index) | 0.68 | 0.62 | 0.40 | 0.59 |
| V (volume) ($m^3$) | 102,900 | 798,300 | 1,106,000 | 2,780,000 |
| $L_{max}$ (maximum length) (m) | 460 | 1282 | 720 | 1320 |
| $W_{max}$ (maximum width) (m) | 252 | 445 | 250 | 570 |
| EI (elongation indicator) | 1.8 | 2.9 | 2.9 | 2.3 |
| Length of shoreline (m) | 1314 | 3163 | 1850 | 4100 |
| SDI (shoreline development index) | 1.3 | 1.4 | 1.4 | 1.5 |

### 2.2. Description of Catchment of the Kartuzy Lakes Complex

The topographic total catchment of the Kartuzy lake complex is covering 12.25 km². The Klasztorna Struga River is the main watercourse of the catchment and depicts the clear hydrographic axis of the area. This is a natural watercourse that is 10.27 km in length, and the starting point is located at the outlet of Lake Mielenko. The Klasztorna Struga River flows through Karczemne, Klasztorne Małe, and Klasztorne Duże. It exits Lake Klasztorne Duże at its northeastern most point. The total catchment area is covered by forests and urbanized areas. Within the total catchment area covering a total of 12.25 km². Their elementary surfaces, the proportion of the total catchment included in the surface and the size of their direct catchments, together with the lake surface area, are presented in Table 2.

**Table 2.** Subcatchments of Klasztorne Duże Lake.

| Partial Catchments of Lakes | Total Catchment [km²] | Proportion in the Catchment of the Complex [%] | Direct Catchment [ha] * | Surface of the Reservoir [ha] |
|---|---|---|---|---|
| MIELENKO | 3.82 | 31.2 | 36.0 | 7.8 |
| KARCZEMNE | 5.15 | 10.8 | 117.6 | 40.4 |
| KLASZTORNE MAŁE | 7.45 | 18.8 | 22.4 | 13.7 |
| KLASZTORNE DUŻE | 12.25 | 39.2 | 161.5 | 57.5 |

*—surface of the topographic catchment without the extent of water drainage system.

Mielenko Lake has a direct catchment area 22.1 ha (65.1% forests, 34.9% meadows). Karczemne Lake has a direct catchment area—44.9 ha (20.7% wastelands, 79.3% forests). Klasztorne Małe Lakes has a direct catchment area 13.0 ha and Klasztorne Duże Lake—103.0 ha (100% forests).

Until 2018, the Kartuzy lake complex and the Klasztorna Struga river received pollution loads from 23 outlets of the rainwater drainage system. Illegally, through the use of several storm collectors, untreated sewage is introduced into the lakes (Karczemne and Klasztorne Małe).

### 2.3. Water Sampling Methods

The physical and chemical properties of the water of studied lakes were conducted in the period from April to November of 2017 and from April to November of 2019. Samples were taken from a depth of 1 m below of surface and from a layer 1 m above the bottom of each lake on site located in the deepest point of lake (Figure 2). In addition, from April to November 2017 and 2019 measurements of the hydrochemical parameters of the water in the surface inflows and outflow were taken (Figure 2).

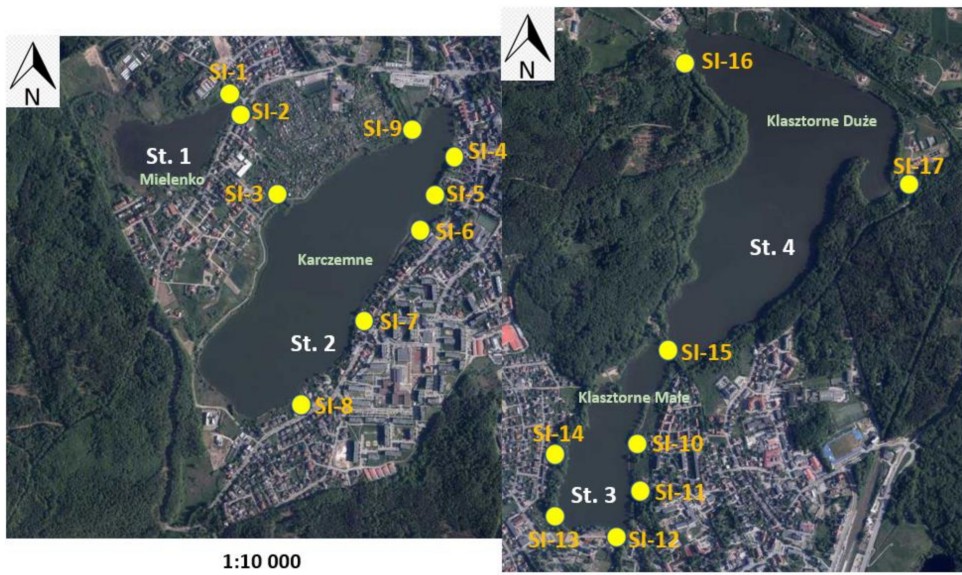

**Figure 2.** Location of sample points.

There were 17 sampling sites, and their detailed characteristics are as follows:

Site 1 (SI-1)—rain collector outlet (Ø 500 mm, a separator);
Site 2 (SI-2)—outflow from Lake Mielenko;
Site 3 (SI-3)—Klasztorna Struga tributary to Lake Karczemne
Site 4 (SI-4)—rain collector outlet (Ø 400 mm, a separator);
Site 5 (SI-5)—rain collector outlet (Ø 600 mm, a separator);
Site 6 (SI-6)—rain collector outlet (Ø 500 mm, a separator);
Site 7 (SI-7)—rain collector outlet (Ø 400 mm, a separator);
Site 8 (SI-8)—rain collector outlet (Ø 500 mm, a separator);
Site 9 (SI-9)—outflow of Klasztorna Struga from Lake Karczemne;
Site 10 (SI-10)—rain collector outlet (Ø 800 mm, without separator);
Site 11 (SI-11)—rain collector outlet (Ø 300 mm, without separator);
Site 12 (SI-12)—Klasztorna Struga tributary to Lake Klasztorne Małe
Site 13 (SI-13)—rain collector outlet (Ø 250 mm, without separator);
Site 14 (SI-14)—rain collector outlet (Ø 800 mm, a separator);
Site 15 (SI-15)—outflow of Klasztorna Struga from Lake Klasztorne Małe;
Site 16 (SI-16)—forest tributary to Lake Klasztorne Duże;
Site 17 (SI-17)—outflow of Klasztorna Struga from Lake Klasztorne Duże.

A VALEPORT 801 device was used to measure the velocity of water, the flow rate was calculated by the Harlacher method [27]. Wherever it was possible to catch the whole water stream flowing into a vessel, a volumetric method was applied to determine the flow intensity [27].

### 2.4. The Physicochemical Analysis of Water

In collected water samples was determined: (total phosphorus) TP, chlorophyll a according to Hermanowicz et al. [28], (total nitrogen) TN (SHIMADZU TOC/TN5000 analyzer), and water transparency by Secchi disc visibility (SD). Every analysis was performed on each triplicate with coefficient of variation according to Kaca [29]. The TP, TN, chlorophyll a, and SD visibility results were statistically analyzed (maximum, minimum, average, Standard deviation, one-way ANOVA, Fisher coefficient F > 1, significance level—$p = 0.05$) [30]. In average values of water parameters were studied significant differences between the year of sewage inflow (2017) and the year after cutting off sewage inflow (2019). The indicators $TSI_{TP}$, $TSI_{TN}$, $TSI_{SD}$, and $TSI_{CHL}$ were calculated according to Carlson [31] and Kratzer and Brezonik [32].

### 2.5. Calculation of External Load of P and N

Nutrient loads at the individual stations were calculated with the generally accepted method of time periods. The external load of the lakes with nutrients originating from surface runoff and with precipitation was calculated with a method that is recommended and applied by the Organisation for Economic Cooperation and Development (OECD) [33–35]. Load of phosphorus (P) and nitrogen (N) introduced with angling baits was calculated according to Wołos et al. [36] and Wołos and Mioduszewska [37]. The permissible and critical phosphorus (P) loadings were calculated according to the hydrological model of Vollenweider [38].

## 3. Results

### 3.1. External Load of Nutrients

The inflow of nutrients to Mielenko Lake from the catchment areas and the atmosphere in 2017 took place via several routes of which the following were of basic importance: areal sources, including direct catchment (3 kg P $y^{-1}$, 72 kg N $y^{-1}$) and water inflow via watercourse flow through forested areas that reached the northeastern part of the lake (1.8 kg P $y^{-1}$, 13 kg N $y^{-1}$); point sources, including rain sewage collected with a separator (2.5 kg P $y^{-1}$, 34 kg N $y^{-1}$) and atmospheric sources (7.5 kg P $y^{-1}$, 30 kg N $y^{-1}$); and recreational sources, including angling biogenic salts introduced with baits (2.7 kg P $y^{-1}$, 55.2 kg N $y^{-1}$) (Figures 3 and 4). The total annual load of phosphorus (P) introduced to the lake was 17.5 kg P $y^{-1}$ (per lake area 0.220 g P $m^2$ $y^{-1}$), whereas that of nitrogen (N) was 204.2 kg N $y^{-1}$ (per lake area 2.600 g N $m^2$ $y^{-1}$).

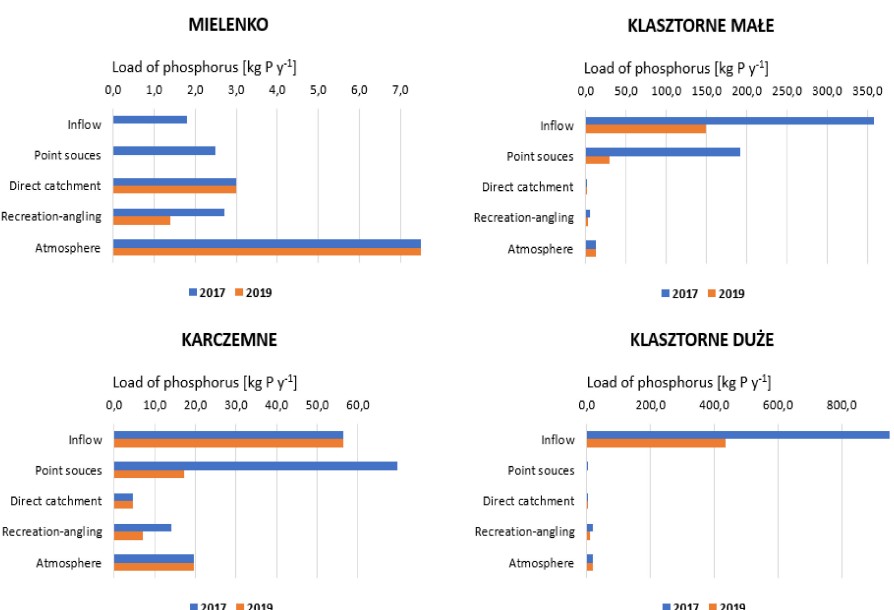

**Figure 3.** Annual load of phosphorus (P) (kg P $y^{-1}$) entering to Kartuzy lake complex.

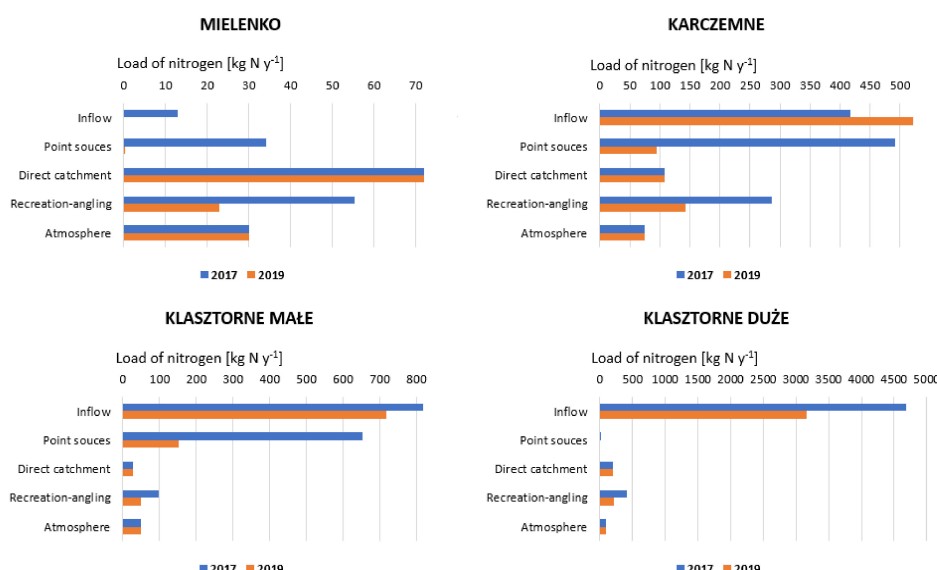

**Figure 4.** Annual load of nitrogen (N) (kg N y$^{-1}$) entering to Kartuzy lake complex.

After modernization of the city's sewage system, the annual load of nutrients introduced to Mielenko Lake by direct catchment (3 kg P y$^{-1}$, 72 kg N y$^{-1}$), point sources (<0.1 kg P y$^{-1}$, 0.1 kg N y$^{-1}$), atmospheric sources (7.5 kg P y$^{-1}$, 30 kg N y$^{-1}$), and recreation-angling biogenic salts introduced with baits (1.4 kg P y$^{-1}$, 23.0 kg N y$^{-1}$) was lowered (Figures 3 and 4). The total annual load of nutrients introduced to the lake in 2019 was 11.9 kg P y$^{-1}$ (per lake area 0.15 g P m$^2$ y$^{-1}$) for phosphorous and 125.1 kg N y$^{-1}$ (per lake area 1.6 g N m$^2$ y$^{-1}$) for nitrogen.

In the case of Karczemne Lake, the inflow of biogenic salts to the lake from the catchment and atmosphere in 2017 involved the following basic components: areal sources (4.5 kg P y$^{-1}$, 108.4 kg P y$^{-1}$), such as the tributary of the Klasztorna Struga river(56.4 kg P y$^{-1}$, 417.5 kg N y$^{-1}$); point sources, including six rain sewage collectors, two with illegal inflow of raw wastewater (69.8 kg P y$^{-1}$, 492.3 kg N y$^{-1}$); atmospheric sources (19.5 kg P y$^{-1}$, 75.2 kg N y$^{-1}$); and recreational sources, such as angling (14.1 kg P y$^{-1}$, 286 kg N y$^{-1}$) (Figures 3 and 4). The total loads of (phosphorus) P and nitrogen (N) introduced to Karczemne Lake in 2017 were 164.3 kg P y$^{-1}$ (per lake area 0.407 g P m$^2$ y$^{-1}$) and 1379.4 kg N y$^{-1}$ (per lake area 3.41 g N m$^2$ y$^{-1}$), respectively. In 2019, after modernization of the sewage system in Kartuzy, the annual load of nutrients was lower: areal sources (4.5 kg P y$^{-1}$, 108.4 kg P y$^{-1}$), such as the tributary of the Klasztorna Struga (56.3 kg P y$^{-1}$, 521.6 kg N y$^{-1}$), point sources, such as that of 4 rain sewage collectors, two with illegal inflows of raw wastewater (17.2 kg P y$^{-1}$, 94.1 kg N y$^{-1}$), atmospheric sources (19.5 kg P y$^{-1}$, 75.2 kg N y$^{-1}$), and recreational sources, such as angling (7 kg P y$^{-1}$, 142 kg N y$^{-1}$) (Figures 3 and 4). The total loads of (phosphorus) P and nitrogen (N) introduced to Karczemne Lake in 2019 were 104.5 kg P y$^{-1}$ (per lake area 0.258 g P m$^2$ y$^{-1}$) and 941.3 kg N y$^{-1}$ (per lake area 2.329 g N m$^2$ y$^{-1}$), respectively.

Klasztorne Małe Lake in 2017 was supplied from a direct catchment (1.3 kg P y$^{-1}$, 26 kg N y$^{-1}$), a surface inflow source (Klasztorna Struga) (358 kg P y$^{-1}$, 818.2 kg N y$^{-1}$), point sources (eight rain sewage collectors, one with illegal inflow of raw wastewater) (192.4 kg P y$^{-1}$, 654 kg N y$^{-1}$), atmospheric sources (12.5 kg P y$^{-1}$, 50.2 kg N y$^{-1}$) and recreational sources (4.8 kg P y$^{-1}$, 97 kg N y$^{-1}$) (Figures 3 and 4). The total load of phosphorus introduced to Klasztorne Małe Lake in 2017 was 569 kg P y$^{-1}$ (per lake area 4.153 g P m$^2$ y$^{-1}$), whereas that of nitrogen was 1654.4 kg N y$^{-1}$ (per lake area 12.076 g N m$^2$ y$^{-1}$). In 2019, with the modernization of the sewage system, nutrient loading to Klasztorne Małe Lake was supplied by direct catchment (1.3 kg P y$^{-1}$, 26 kg N y$^{-1}$), surface inflow (Klasztorna Struga) (149 kg P y$^{-1}$, 718 kg N y$^{-1}$), point sources (one rain sewage collector) (29.3 kg P y$^{-1}$, 151.3 kg N y$^{-1}$), atmospheric sources (12.5 kg P y$^{-1}$, 50.2 kg N

$y^{-1}$) and recreational sources (2.5 kg P $y^{-1}$, 48 kg N $y^{-1}$) (Figures 3 and 4). The total load of phosphorus introduced to Klasztorne Małe Lake in 2019 was 194.6 kg P $y^{-1}$ (per lake area 1.420 g P $m^2$ $y^{-1}$), whereas that of nitrogen was 993.5 kg N $y^{-1}$ (per lake area 7.252 g N $m^2$ $y^{-1}$).

The recharge of Klasztorne Duże Lake with nutrients originated from areal sources (1.3 kg P $y^{-1}$, 206 kg N $y^{-1}$), inflows from Klasztorne Małe Lake and the surrounding forest tributaries (952 kg P $y^{-1}$, 4680 kg N $y^{-1}$), point sources (0.3 kg P $y^{-1}$, 4 kg N $y^{-1}$), atmospheric sources (20 kg P $y^{-1}$, 95 kg N $y^{-1}$) and recreational sources (20.1 kg P $y^{-1}$, 407.1 kg N $y^{-1}$) (Figures 3 and 4). The total phosphorus load introduced to Klasztorne Duże Lake in 2017 was 993.7 kg P $y^{-1}$ (per lake area 1.755 g P $m^2$ $y^{-1}$), and that of nitrogen was 5392.1 kg N $y^{-1}$ (per lake area 9.526 g N $m^2$ $y^{-1}$). In 2019, after protective treatments were implemented, the external load of nutrients was lower: areal sources (1.3 kg P $y^{-1}$, 206 kg N $y^{-1}$), inflow from Klasztorne Małe Lake and the surrounding forest tributaries (434.7 kg P $y^{-1}$, 3162.8 kg N $y^{-1}$), point sources (<0.1 kg P $y^{-1}$, 0.3 kg N $y^{-1}$), atmospheric sources (20 kg P $y^{-1}$, 95 kg N $y^{-1}$) and recreational sources (10 kg P $y^{-1}$, 215 kg N $y^{-1}$) (Figures 3 and 4). The total phosphorus load introduced to Klasztorne Duże Lake in 2019 was 466 kg P $y^{-1}$ (per lake area 0.823 g P $m^2$ $y^{-1}$), and the total nitrogen load was 3678.8 kg N $y^{-1}$ (per lake area 6.499 g N $m^2$ $y^{-1}$).

### 3.2. Water Chemistry

The cutoff sewage inflow to studied lakes caused highly significant variation in the total phosphorus (TP) only in the bottom water layers of the Karczemne Lake and the entire water volume of the Klasztorne Małe Lake (Table 3). In 2017, the mean TP concentration in the surface water layer of Mielenko Lake was 0.189 mg P $L^{-1}$ ($\pm$0.079) and 0.277 mg P $L^{-1}$ ($\pm$0.122) in the near-bottom layer (Figure 5, Table 4). In 2019, after protective treatments were implemented, the average TP concentration of the entire volume of water was 0.185 mg P $L^{-1}$ ($\pm$0.059).

**Table 3.** Results of statistical analyses (F-Fisher coefficient, *p*-significance level).

| Variable | Mielenko | | Karczemne | | Klasztorne Małe | | Klasztorne Duże | |
|---|---|---|---|---|---|---|---|---|
| | **F** | ***p*** | **F** | ***p*** | **F** | ***p*** | **F** | ***p*** |
| SD visibility | 0.00000 | 1.00000 | 0.162791 | 0.692695 | 0.144909 | 0.709162 | 0.081733 | 0.779149 |
| Chlorophyll a | 0.589456 | 0.455385 | 0.372783 | 0.551275 | 0.022063 | 0.884037 | 4.418058 | 0.054141 |
| TP surface | 0.008555 | 0.927616 | 0.883513 | 0.363177 | 9.547787 | 0.007990 | 1.372432 | 0.260947 |
| TP bottom | 3.633657 | 0.077361 | 5.616239 | 0,032701 | 9.426891 | 0.008308 | 0.505114 | 0.488933 |
| TN surface | 3.28202 | 0.091544 | 0.00001 | 0.997200 | 2.29978 | 0.151644 | 2.50773 | 0.135611 |
| TN bottom | 2.51353 | 0.135195 | 0.36897 | 0.553293 | 39.72779 | 0.00001 | 0.16379 | 0.691807 |

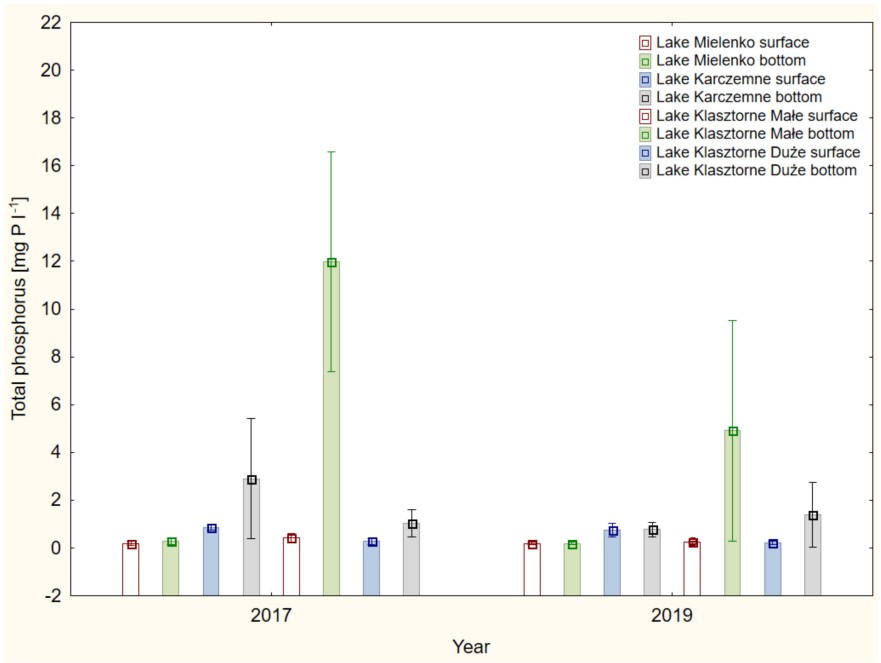

**Figure 5.** Mean annual values of total phosphorus (TP) in the water of Kartuzy lake complex in selected research years (box ± SEM, whisker plots ± Standard Deviation).

**Table 4.** Variability of nutrients, chlorophyll a and Secchi disc visibility in study lakes.

| | | Mielenko | | Karczemne | | Klasztorne Małe | | Klasztorne Duże | |
|---|---|---|---|---|---|---|---|---|---|
| | | **2017** | | | | | | | |
| | | Min/Max | SD | Min/Max | SD | Min/Max | SD | Min/Max | SD |
| TP | surface | 0.119/0.368 | 0.079 | 0.613/0.950 | 0.120 | 0.251/0.780 | 0.179 | 0.134/0.446 | 0.118 |
| [mg L⁻¹] | bottom | 0.147/0.490 | 0.122 | 0.667/7.460 | 2.522 | 7.760/20.60 | 4.598 | 0.295/1.930 | 0.566 |
| TN | surface | 1.47/3.01 | 0.541 | 1.32/5.57 | 1.320 | 1.83/2.40 | 0.182 | 1.25/2.12 | 0.273 |
| [mg L⁻¹] | bottom | 1.43/3.00 | 0.531 | 1.33/4.71 | 1.008 | 17.24/22.38 | 2.306 | 1.87/6.34 | 1.565 |
| Chlorophyll a [µg L⁻¹] | | 8.4/43.8 | 13.88 | 26.6/125.1 | 32.60 | 28.6/64.5 | 12.65 | 15.6/70.1 | 16.81 |
| SD Visibility [m] | | 0.4/0.8 | 0.138 | 0.2/0.5 | 0.112 | 0.3/1.1 | 0.267 | 0.4/1.8 | 0.467 |
| | | **2019** | | | | | | | |
| | | Min/Max | SD | Min/Max | SD | Min/Max | SD | Min/Max | SD |
| TP | surface | 0.109/0.273 | 0.059 | 0.360/1.222 | 0.298 | 0.165/0.301 | 0.044 | 0.119/0.296 | 0.066 |
| [mg L⁻¹] | bottom | 0.109/0.273 | 0.059 | 0.381/1.278 | 0.303 | 2.028/15.62 | 4.611 | 0.204/4.000 | 1.367 |
| TN [mg | surface | 1.42/5.52 | 1.297 | 2.27/5.48 | 1.042 | 1.72/2.61 | 0.314 | 1.51/2.90 | 0.449 |
| L⁻¹] | bottom | 1.42/5.52 | 1.297 | 2.34/5.52 | 0.998 | 24.39/43.32 | 7.047 | 2.02/8.94 | 2.319 |
| Chlorophyll a [µgL⁻¹] | | 11.5/65.4 | 19.38 | 14.4/170.9 | 50.7 | 13.1/76.2 | 24.27 | 7.0/45.4 | 14.53 |
| SD Visibility [m] | | 0.2/1.1 | 0.282 | 0.2/0.35 | 0.067 | 0.3/0.9 | 0.185 | 0.5/2.0 | 0.494 |

In 2017, the mean TP concentration in the surface water layer of Karczemne Lake was 0.853 mg P L$^{-1}$ (±0.120, and 2.899 mg P L$^{-1}$ (±2.522) in the near-bottom layer. In 2019, after modernization of the city's sewage system, the average TP concentration in the surface water layer of Karczemne was 0.746 mg P L$^{-1}$ (±0.298), and the near-bottom TP concentration was 0.771 mg P L$^{-1}$ (±0.303) (Figure 5, Table 4). In Klasztorne Małe Lake, in 2017, the mean TP concentration in the surface water layer was 0.434 mg P L$^{-1}$ (±0.179), and was 11.976 mg P L$^{-1}$ (±4.598) at the lake bottom (Figure 5, Table 4). In 2019, in the meromictic Klasztorne Małe Lake, the mean TP concentration in the surface water layer was 0.232 mg P L$^{-1}$ (±0.044) and 4.907 mg P L$^{-1}$ (±4.611) in the bottom layer (Table 4).

In 2017, in the case of Klasztorne Duże Lake, the mean content of TP in the surface water layer was 0.275 mg P L$^{-1}$ (±0.118), and in the near-bottom water, it was 1.024 mg P L$^{-1}$ (±0.566). In 2019, after protective techniques were implemented, the mean concentration of TP was 0.219 mg P L$^{-1}$ (±0.066) in the surface water of Klasztorne Duże Lake, while that at the bottom was 1.396 mg P L$^{-1}$ (±1.367) (Figure 5, Table 4).

The cutoff of the sewage inflow to the lakes caused highly significant variability in the total nitrogen (TN) values only in the bottom water layers of Klasztorne Małe Lake (Table 3). In the surface water layer of Mielenko Lake, in 2017, the average concentration of TN was 2.086 mg N $l^{-1}$ ($\pm0.185$), and in the near-bottom water, it was 2.200 mg N $l^{-1}$ ($\pm0.531$) (Figure 6, Table 4). In 2019, after protective treatments were implemented, in the entire water volume of Mielenko Lake, the mean concentration of TN was 2.986 mg N $l^{-1}$ ($\pm1.297$). In 2017, the mean total nitrogen concentration in the surface water layer of Karczemne Lake was 3.191 mg N $l^{-1}$ ($\pm1.320$), and near the bottom, it was 2.975 mg N $l^{-1}$ ($\pm1.008$). In 2019, in the surface water layer of the lake, the mean concentration of TN was 3.188 mg N $l^{-1}$ ($\pm1.042$), and the near-bottom content was 3.280 mg N $l^{-1}$ ($\pm0.998$) (Figure 6, Table 4). In 2017, in Klasztorne Małe Lake, the average TN concentration in the surface water layer was 1.740 mg N $l^{-1}$ ($\pm0.182$), while at the bottom, it was 19.450 mg N $l^{-1}$ ($\pm2.306$) (Table 4). In 2019, the average total nitrogen concentration in the water of the lake was 2.177 mg N $l^{-1}$ ($\pm0.314$), while at the bottom, it was 35.973 mg N $l^{-1}$ ($\pm7.047$) (Table 4).

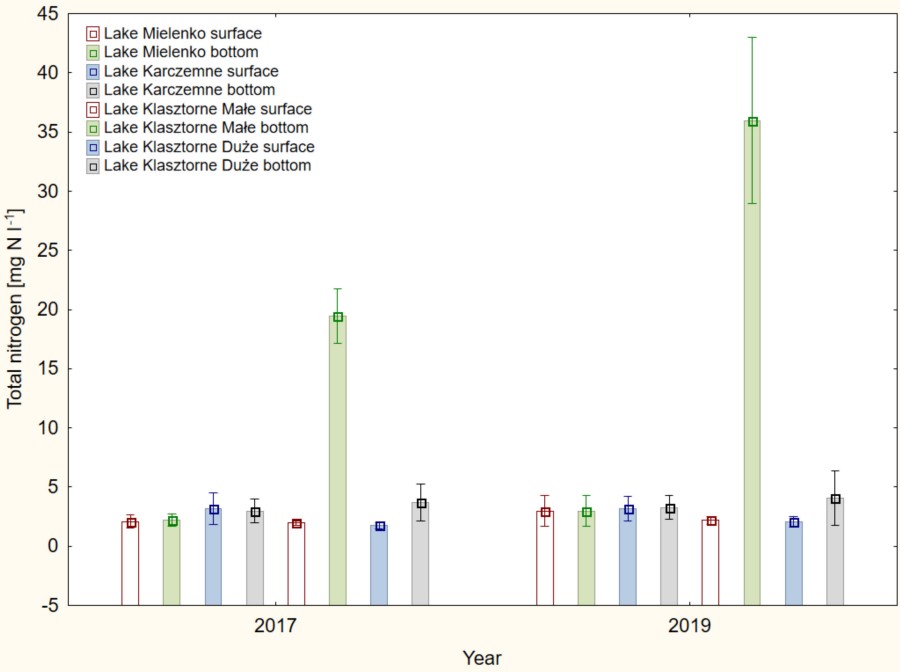

**Figure 6.** Mean annual values of total nitrogen (TN) in the water of Kartuzy lake complex in selected research years (box $\pm$ SEM, whisker plots $\pm$ Standard Deviation).

In the surface water layer of Klasztorne Duże Lake, in 2017, the mean content of TN was 1.740 mg N $l^{-1}$ ($\pm0.273$), and that of the near-bottom water layer was 3.664 mg N $l^{-1}$ ($\pm1.565$) (Figure 6, Table 4). In 2019, after modernization of the city's sewage system, the mean concentration of TN in the surface water layer of Klasztorne Duże Lake was 2.035 mg N $l^{-1}$ ($\pm0.449$), while at the bottom, it was 4.065 mg N $l^{-1}$ ($\pm2.319$).

The cutting off sewage inflow to lakes did not cause significant variability in the chlorophyll a content and SD visibility values in the lakes (Table 3). In 2017, the average chlorophyll a level in the water of Mielenko Lake was 28.9 μg $l^{-1}$ ($\pm13.88$), and in 2019, after protective treatments were implemented, it was 35.4 μg $l^{-1}$ ($\pm19.38$) (Figure 7, Table 4). In the water of Karczemne Lake, the mean content of chlorophyll a in 2017 was 90.4 μg $l^{-1}$ ($\pm32.60$), while in 2019, it was 77.4 μg $l^{-1}$ ($\pm50.70$) (Figure 7, Table 4). In 2017, in Klasztorne Małe Lake, the average chlorophyll a level was 45.3 μg $l^{-1}$ ($\pm12.65$), and in 2019, it was 43.8 μg $l^{-1}$ ($\pm24.27$) (Table 4). In 2017, the average chlorophyll a content in the water of Klasztorne Duże Lake was 42.1 μg $l^{-1}$ ($\pm16.81$), while in 2019, it was 25.6 μg $l^{-1}$ ($\pm14.53$) (Figure 7, Table 4).

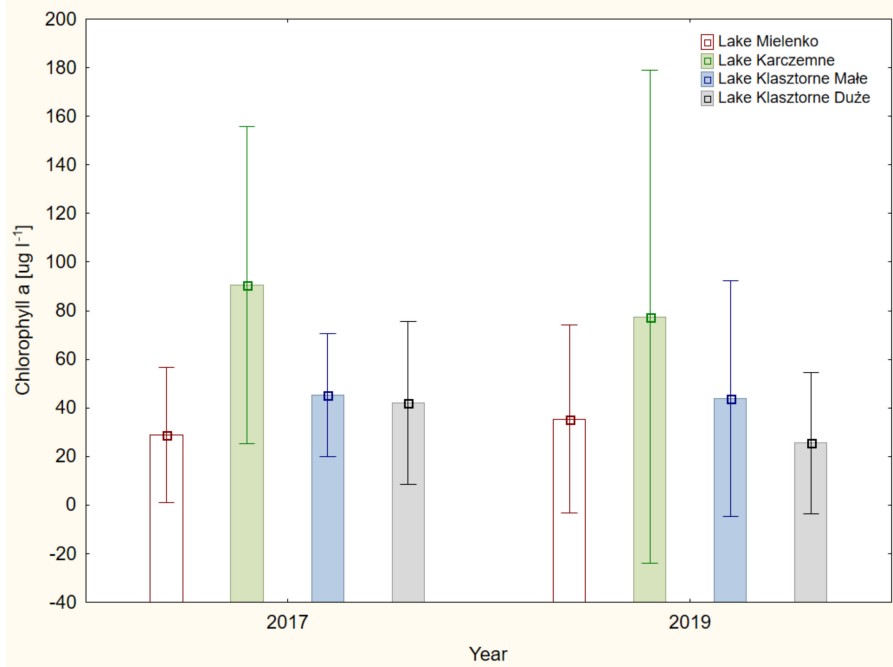

**Figure 7.** Mean annual values of chlorophyll a in the water of Kartuzy lake complex in selected research years (box ± SEM, whisker plots ± Standard Deviation).

The mean value of Secchi disc visibility (SD) in 2017 in Mielenko Lake was 0.525 m (±0.138), and in 2019, it was also 0.525 m, but with standard deviations of -±0.282 (Figure 8, Table 3). In 2017, the average Secchi disc visibility in Karczemne Lake was 0.312 m (±0.112), and in 2019, it was 0.293 (±0.067) (Figure 8, Table 4). In Klasztorne Małe Lake, in 2017, the mean Secchi disc visibility was 0.575 m (±0.267), while in 2019, after modernization of the city's sewage system, it was 0.531 m (±0.185) (Figure 8, Table 4). In 2017, the average Secchi disc visibility in Klasztorne Duże Lake was 0.812 m (±0.467), and in 2019, it was 0.881 m (±0.494) (Figure 8, Table 4).

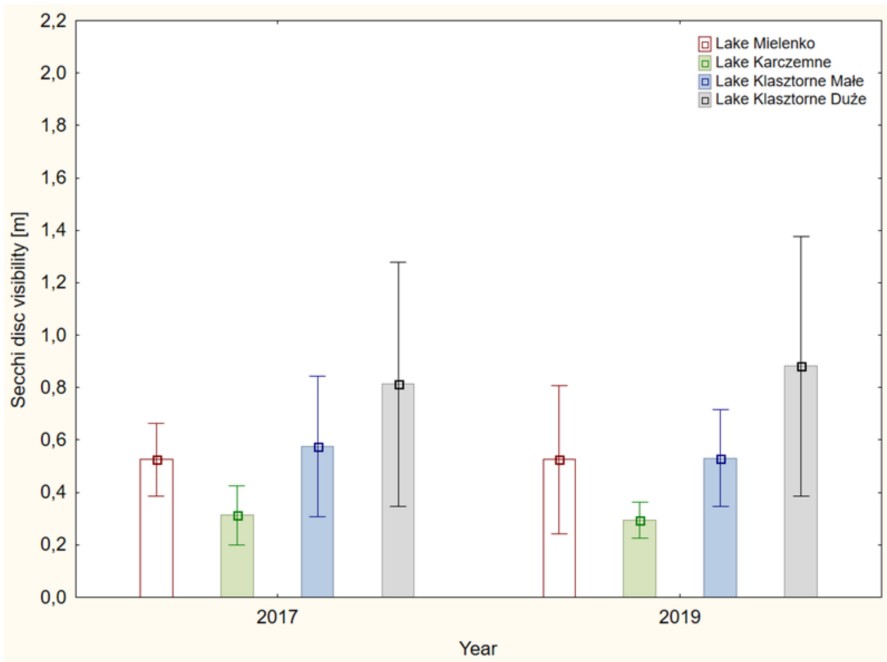

**Figure 8.** Mean annual values of transparency of Kartuzy lake complex in selected research years. (box ± SEM, whisker plots ± Standard Deviation)

## 4. Discussion

Freshwater availability is a major issue across the world [39]. Approximately one-third of the required drinking water is obtained from surface sources, such as rivers, lakes or dams and canals. These sources of water also serve as the best receivers for the discharge of stormwater and domestic or industrial wastes. These conditions are very dangerous due to the constant development of agricultural and urbanized areas and the related increase in pollutant loads of different origins introduced into the waters [40,41]. Among these point sources, untreated municipal wastewater has been identified as the most hazardous to water ecosystems due to its large amount of nutrients and high organic matter content [42]. Raw domestic sewage also includes pathogenic organisms, oxygen-demanding organic substances and inorganic and organic toxic substances [43,44].

Since 1956, the Kartuzy Lake complex, which includes the Mielenko, Karczemne, Klasztone Małe, and Klasztorne Lakes, has been used as a receiver for domestic sewage sources, including that from dairy farms, slaughterhouses, breweries, furniture factories, and municipal hospitals. These lakes receive an excessive load of nutrients.

In 2017, the Kartuzy Lakes and Klasztorna Struga River received sewage from 23 runoff collectors. Although some of the outlets were equipped with separators, this does not provide sufficient protection against the inflow of nitrogen and phosphorus, which are the elements responsible for eutrophication. Biogenic substances are mainly removed in their insoluble form together with mineral suspensions (predominantly phosphorus) and organic fractions (mainly nitrogen). Unfortunately, it is the dissolved forms of these substances, such as phosphates, nitrite nitrogen and ammonium ions, that provide an excellent medium for phytoplankton that cannot be entirely eliminated [45,46].

Considering the proportion of individual sources of contamination in the context of phosphorus supplies, the biggest hazard to water quality in the lakes were as follows: angling in Mielenko Lake (ca. 45%); point sources (ca. 42%) and the Lake Mielenko tributary (ca. 34%) for Karczemne Lake; the Klasztorna Struga Lake tributary (ca. 60.4%) and point sources (ca. 36%) for Klasztorne Małe Lake; and the Klasztorne Małe Lake tributary (ca. 95%) for Lake Klasztorne Duże.

The annual load of phosphorus sourced from the catchment and atmosphere, considering the permissible and dangerous concentrations calculated from Vollenweider's hydrological model [38], were compared. The analysis revealed that the total phosphorus concentration that was introduced from external sources to the individual lakes in 2017 substantially exceeded the dangerous concentrations that are known to cause accelerated eutrophication. The annual phosphorus concentration that was introduced to Lake Mielenko in 2017 was almost 200% higher than the critical concentration; in Lake Karczemne, it was six times higher (670% in relation to the critical concentration); in Klasztorne Małe, it was seven times higher (770% in relation to the critical concentration); and in Lake Klasztorne Duże, it was 1000% higher (nine times higher than the critical load). A strategy for the protection of the Kartuzy lake complex was developed. The protective measures were implemented to limit the amount of fishing bait introduced into the lakes and to limit the inflow of sewage from rain collectors. In 2018, the Municipal Water and Sewage Company in Kartuzy modernized the city's sewage network. The modernization involved, among other things, the replacement of piping, the construction of reservoirs for excess water, and the reconstruction of the sewage pumping station.

These protective measures reduced the external loading of nutrients into the lakes. In the case of Mielenko Lake, which was the lake with the least pollutant loading from the outlets of the sewage collectors, after the modernization of the city's sewage system, a 37% decrease in the external phosphorus load and a 32% decrease in the external nitrogen load were observed. In Karczemne Lake, a 32% decrease in phosphorus load and a 39% decrease in external nitrogen load were noted. In the case of Klasztorne Małe Lake, up to a 66% decrease in phosphorus external load and a 40% decrease in nitrogen load were found. In Klasztorne Duże Lake, the external loads of nutrients (P and N) decreased by 54% and 32%, respectively.

The multiannual discharge of raw domestic- and storm-derived sewage into lakes resulted in an extremely high concentration of phosphorus compounds, similar to those found in diluted wastewater. The total phosphorus (TP) concentration in the bottom layer of the waters of Klasztorne Małe Lake exceeded 20 mg P $L^{-1}$. Such high values were not recorded in any other lake degraded by sewage [47–50]. In a moderately eutrophic lake, the phosphorus cycle is regulated mainly by phytoplankton, and during its maximal developmental stage, the phosphate level may decrease to analytical zero. In the water of the Kartuzy lake complex, despite its extremely high primary production (a strong oxygen oversaturation, a high pH above 9.5, a water transparency of 0.2–0.3 m, and a high organic matter content), the total depletion of phosphorus in the water was not noted. The concentrations here were so high that they were in excess of the amount that phytoplankton could consume. After severing the sewage inflow, these conditions did not change significantly. According to Berleć et al. [51] and Sondengaard et al. [22], it is certainly related to phosphorus release from bottom sediments. This process took place even in the shallow lakes of Mielenko and Karczemne because the bottom sediments of these lakes are rich in phosphorus. Although the range of release from bottom sediment under aerobic conditions is generally lower than that in anaerobic conditions, nutrients are introduced directly into the trophogenic zone, where they are used by phytoplankton [52,53]. The extremely high concentration of phosphorus in the near-bottom layer of Klasztorne Małe Lake was related to the permanent anoxic conditions in the over-bottom water and thus restricted P binding by bottom sediment. Under low redox potential conditions, phosphates released by desorption and in the organic matter destruction processes in the bottom sediment diffuse toward the over-bottom water layer [54]. The discharge of raw municipal sewage into the study lakes resulted in extremely high concentrations of nitrogen. The average total nitrogen (TN) concentration reached 43 mg N $l^{-1}$. After the sewage inflow was cut off, these conditions did not change. Hamersly et al. [55] noted that nitrogen depletion in water can be caused by the increased sedimentation of organic matter and its deposition in sediment or by the ammonification and denitrification of organic compounds to free nitrogen. It should be emphasized that the formation of free nitrogen is only possible after prior nitrification, which was possible in the Mielenko and Karczemne Lakes. Microorganisms play a major role in nitrogen transformation, and their activity is dependent on many environmental factors. Ammonification may occur under both aerobic and anaerobic conditions over a wide range of pH values and temperatures, and the intensity of these processes depends on the number of respective physiological groups of bacteria [56]. According to Höhener and Gächter [57], increased temperatures favor the release of ammonia nitrogen from sediment to water. In the Klasztorne Małe and Klasztorne Duże Lakes, where the bottom water layer was anaerobic, ammonia nitrogen was present in very high concentrations and was the dominant nitrogen compound form. Sufficient oxygen conditions in the deeper parts of the Mielenko and Karczemne Lakes resulted in a decrease in the ammonia nitrogen content by oxidizing to nitrate nitrogen. The processes of ammonia oxidation (nitrification) are favored under elevated temperatures, which was achieved in the polymictic lakes of Mielenko and Karczemne by water mixing and heating via contact with the atmosphere. Ammonia in the surface layer of lake water can be consumed by intensively developing phytoplankton, and in the bottom, sediment can be reduced to gaseous nitrogen (denitrification). According to Helmroos et al. [58] and Ok-Sun et al. [59], denitrification is the main process leading to the loss of nitrogen from aquatic ecosystems. The water pH levels also influence the intensity of this phenomenon, and the optimum ranges is between pH 7.0 and pH 8.2. However, the most important factor is proper oxygen conditions [60].

The trophic conditions in the Kartuzy lake complex were determined in the year preceding the implementations of protective measures in the catchment area and after the modernization of the city's sewage system. The trophic statuses of the lakes were assessed as the result of causative factors (phosphorus and nitrogen concentration) and effect-causal factors (visibility of the Secchi disc and chlorophyll a concentration) during the

growing season. The Kartuzy lake complex, as an urban reservoir and a sewage receiver, was intensively exposed to anthropogenic pressures and strongly degraded. Before the protective measures were implemented and after the modernization of the city's sewage system, the relationship between the basic trophic indices was constant, TSI (TP) > TSI (SD) > TSI (Chl a), and the values were in the ranges characteristic of eutrophication (58-82) (Figure 8). The indicator for TP was the only pollutant that fell within the range of hypertrophy (Figure 9), which is a common phenomenon in many lakes in Poland. The $TSI_{TP}$ often calculates higher values than those indicated by the actual state of the environment [61,62]. However, this did not apply to the lakes in question, where the visibility of SD was characteristic of highly eutrophic reservoirs [63] and limited to 0.20–0.55 m, where TSI (SD) > TSI (Chl a). This indicates the presence of small algae that limit the visibility [64]. In Klasztorne Małe Lake, the chlorophyll a concentrations in July 2019 reached 72.77 µg l$^{-1}$, with a visibility SD = 0.5 (one of the highest values obtained), and in Karczemne Lake, the chlorophyll a concentrations were much higher, reaching 170.92 µg l$^{-1}$ with an SD visibility of 0.25 m. For all lakes, no correlation was found between the visibility of SD and the concentration of chlorophyll a, which confirms the above theory. The limiting role of nitrogen was demonstrated, but it was not evidenced by the relationships of TSI (Chl a)—TSI (TN) < 0 and TSI (TN)—TSI (TP) < 0 [32].

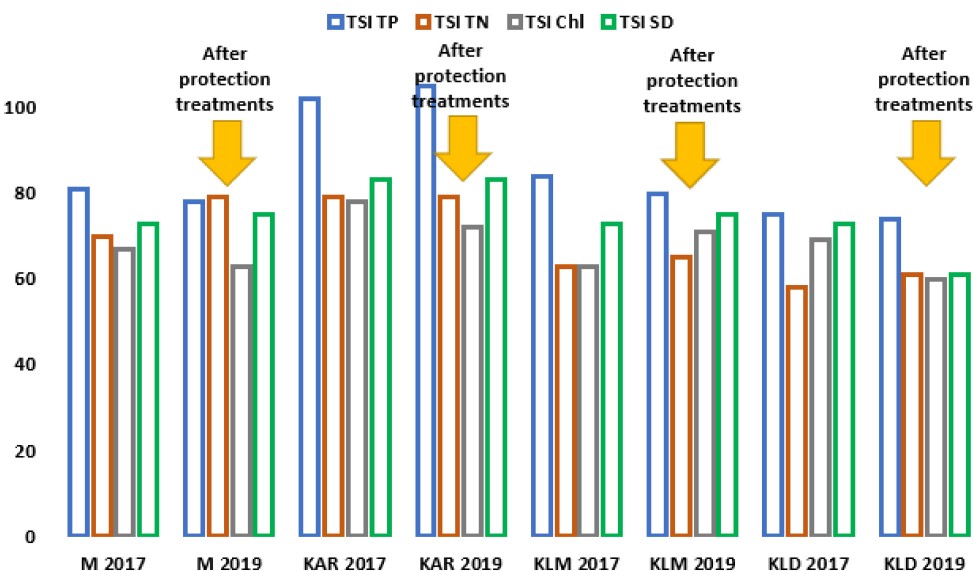

**Figure 9.** The Trophic Status Index (TSI) calculated on the basis of visibility (SD) chlorophyll concentrations (ChL), total phosphorus (TP) and total nitrogen (TN).

## 5. Conclusions

Research preformed in the Kartuzy Lake complex demonstrated the deep degradation of water quality in these reservoirs. Long-term sewage inflow has influenced the maintenance of the very high nutrient concentrations present in the water and bottom sediments of these lakes. Even after severing the flow of pollutants into the lakes, they were able to maintain a very high eutrophic status. The implementation of the protective measures resulted in a slight decrease in the concentrations of phosphorus and nitrogen in the water. However, these changes did not improve the environmental conditions of the lakes. In a situation where the internal fertilization process in the lakes has begun, the improvement of water quality is only possible via restoration efforts with methods that have been adjusted to the individual characteristics of each lake.

The presented research results clearly show that it is very important to prevent lake pollution, because the effects of degradation are sometimes irreversible.

**Author Contributions:** Conceptualization, J.G. methodology, J.G., R.T. investigation, J.G., R.Twriting—original draft preparation, J.G., R.T.; writing—review and editing, J.G., R.T.; visualization, J.G.,

R.T.; funding acquisition, R.T. All authors have read and agreed to the published version of the manuscript."

**Funding:** This research was funded by COMMUNITY OF KARTUZY, 29.690.048-500.

**Institutional Review Board Statement:** Not Applicable.

**Informed Consent Statement:** Not applicable.

**Data Availability Statement:** No new data were created or analyzed in this study. Data sharing is not applicable to this article.

**Acknowledgments:** Project financially supported by Minister of Science and Higher Educationin the range of the program entitled "Regional Iniciative of Excellence" for the years 2019–2022, project No. 010/RID/2018/19, amount funding 12.000.000 PLN.

**Conflicts of Interest:** The authors declare no conflict of interest. The funders had no role in the design of the study; in the collection, analyses, or interpretation of data; in the writing of the manuscript, or in the decision to publish the results.

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
