# Peer review of "The Influence of the Modernization of the City Sewage System on the External Load and Trophic State of the Kartuzy Lake Complex"

_applsci, doi:10.3390/app11030974_

Round 1
Reviewer 1 Report
The case-study presented in this paper is the trophic evolution of the reservoirs in the Kartuzy lakes complex and its connection with the sewer management in the drained catchment. In particular, the Authors analyse the effects of some restoring actions on the quality status of the lakes.
The subject is of interest of the scientific community. Even though a large portion of the punctual nutrient sources in lakes was limited by the sewages collections, as well as by the operation of waste water treatment plants, the interaction between combined sewer systems and lake is still a challenge, especially in wet period when the overflows activate. Moreover, the case presented by the Authors is quite interesting due the extreme state of degradation of the involved water bodies, and has thus the potential for a research paper.
Though, the actual version of the work is quite far from a research paper. The main weaknesses that I see are the following ones:
- The study is presented as a technical report on a specific case-study instead of a research paper. Starting from the Instruction, the reader does not understand what is the research question and which is the novelty of the work. The topic presented in the Introduction is very generalist and no focus on a specific issue is present. The results section is full of numbers which simply describe in words what is already presented in the picture. No state-of-the-art methodology is used to interpret the data. No discussion of the research findings with respect to other research works is present.
- Even though I am not a mother tongue, I think that the level of English is very poor and often there are examples of wrong expressions. Often I observed the lack of correct technical words. Examples: “phosphorus charge” in place of phosphorus load, “the dangerous load”, “method of land management”, “Secchi disc visibility” …
- In general , I had the impression of a insufficient care in the paper writing. There are errors that would be immediately detached after a quick reading, for example sentences like: “In the case of Klasztorne MaÅ‚e Lake, as much as 66% decrease in phosphorus external load and 40% decrease in nitrogen load was found.40% decrease in external load of nitrogen was found.” Or “The protective measures reduced the external load of nutrients. In case of Mielenko Lake annual load of phosphorus. In”. Another example is the references: in the Introduction, two different way of citing were used (with year/ with the number)
Author Response
Answer to Reviewer 1
Dear Reviewer
I would like to thank You very much for the review. I think that Your comments are very accurate and necessary to improve the quality of the manuscript. I made all your remarks in the text. I hope that the changes introduced by me in accordance with Your recommendations will be sufficient for the manuscript to be accepted for publication.
I agree that the problem of transforming lakes into recipients of sewage is still present. As suggested by the Reviewer, I extended the introduction to the manuscript so that the research question and the innovative nature of the study could be read. I changed the description of the results and added a statistical study. I eliminated carelessly prepared elements of the work, standardized the quotation of literature. The manuscript has been revised by Native Speaker.
Reviewer 2 Report
Comments on Manuscript titled “THE INFLUENCE OF CITY'S SEWAGE SYSTEM MODERNIZATION ON THE EXTERNAL LOAD AND THE TROPHIC STATE OF THE LAKE COMPLEX”
- Abstract seems lengthy and abstracts are generally single paragraph thing.
- LINE 4 OF abstract words are joined “their waters.In 2018”. I would suggest to check everything rigorously.
- The first para of abstract could be said in just one or two line. Shorten it.
- The last paragraph of introduction should identify the after impacts of such a study. It could be a one line explanation telling how the results of this study will benefit society or academician working on this problem.
- The paper seems a little lengthy. Similar information in results section and tables/figures could be avoided to cut it down.
Author Response
Answer to Reviewer 2
Dear Reviewer
I would like to thank You very much for the review. I think that Your comments are very accurate and necessary to improve the quality of the manuscript. I made all your remarks in the text. I hope that the changes introduced by me in accordance with Your recommendations will be sufficient for the manuscript to be accepted for publication.
As suggested by the Reviewer, I shortened the manuscript abstract, changed the description of the results and added a statistical study. I eliminated carelessly prepared elements. The manuscript has been revised by Native Speaker.
Reviewer 3 Report
At its current form, there is not any intriguing and exciting conclusion in the study. The research work draws the predictable conclusion of high nutrient concentrations in the water and bottom sediments of these lakes with urbanization which is not an unknown finding.
Also, the paper at its current state appears like a technical report rather than a journal paper. The authors mention the aim of the study is to present the impact of protective measures consisting in the modernization of the city's sewage system on the quality of water in lakes and their trophic state, but the paper currently appears as the report of collected data and there is no depth analysis to the collected information. Thus I do not see other readers being benefitted by the information provided in the paper, unless it’s the people living in the periphery of the Kartuzy lakes complex. The authors mention “Every analysis was performed in triplicate” yet standard deviation is not provided in any of the fig3 or fig4 or fig5. The paper need reorganization and reshuffling so that researchers around the world could use the findings from this work in their context.
Author Response
Answer to Reviewer 3
Dear Reviewer
I would like to thank You very much for the review. I think that Your comments are very accurate and necessary to improve the quality of the manuscript. I made all your remarks in the text. I hope that the changes introduced by me in accordance with Your recommendations will be sufficient for the manuscript to be accepted for publication.
As suggested by the Reviewer, I changed the work so that it had a scientific character and its results could be valuable to scientists all over the world. The manuscript has been revised by Native Speaker.
Round 2
Reviewer 3 Report
None
Author Response
Dear Reviewer
I would like to thank You very much for the second review. I think that Your comments are very accurate and necessary to improve the quality of the manuscript. I made all your remarks in the text – I corrected minor language mistakes. I hope that the changes introduced by me in accordance with Your recommendations will be sufficient for the manuscript to be accepted for publication.
Sincerely Yours
Jolanta Grochowska